# Synergistically Boosting Li Storage Performance of MnWO_4_ Nanorods Anode via Carbon Coating and Additives

**DOI:** 10.3390/ma17194682

**Published:** 2024-09-24

**Authors:** Duo Wang, Zhaomin Wang, Chunli Wang, Dongming Yin, Yao Liang, Limin Wang, Yong Cheng, Ming Feng

**Affiliations:** 1Key Laboratory of Functional Materials Physics and Chemistry of Ministry of Education, Jilin Normal University, Changchun 130103, China; dwang@jlnu.edu.cn; 2State Key Laboratory of Rare Earth Resource Utilization, Changchun Institute of Applied Chemistry, Chinese Academy of Sciences, Changchun 130022, China; zmwang@ciac.ac.cn (Z.W.); clwang@ciac.ac.cn (C.W.); dmyin@ciac.ac.cn (D.Y.); liangyao@ciac.ac.cn (Y.L.); lmwang@ciac.ac.cn (L.W.)

**Keywords:** MnWO_4_@C, nanocavities, 3D frameworks, pseudo-capacitance, electrolyte additive

## Abstract

Polyanionic structures, (MO_4_)^n−^_,_ can be beneficial to the transport of lithium ions by virtue of the open three-dimensional frame structure. However, an unstable interface suppresses the life of the (MO_4_)^n−^-based anode. In this study, MnWO_4_@C nanorods with dense nanocavities have been synthesized through a hydrothermal route, followed by a chemical deposition method. As a result, the MnWO_4_@C anode exhibits better cycle and rate performance than MnWO_4_ as a Li-ion battery; the capacity is maintained at 718 mAh g^−1^ at 1000 mA g^−1^ after 400 cycles because the transport of lithium ions and the contribution of pseudo-capacitance are increased. Generally, benefiting from the carbon shell and electrolyte additive (e.g., FEC), the cycle performance of the MnWO_4_@C electrode is also effectively improved for lithium storage.

## 1. Introduction

In recent years, due to the theoretical capacity of commercial graphite negative electrodes being only 372 mAh g^−1^, they are no longer able to meet the requirements of the new generation of lithium-ion batteries as high energy density and high power density negative electrodes. Therefore, scientists are exploring a new type of negative electrode material with high specific energy and high energy density to replace graphite negative electrodes [1,2,3,4].

Metal tungstates (MWO_4_, where M is a 3D divalent transition metal ion with an ionic radius < 1 A°) with the wolframite type of structure can be described as made up of hexagonal close-packed oxygen atoms with certain octahedral sites filled with M^2+^ and W^6+^ cations in an ordered way [5,6]. A large amount of research has been conducted on this series of compounds in many application fields, but there is limited literature on the study of metal tungstate salts in the field of electrochemistry [7,8,9,10,11,12,13,14,15]. Fortunately, transition metal oxides have advantages, such as high specific capacity, lower operating voltage, and environmental friendliness, and are considered one of the most promising negative electrode materials. In addition, their preparation cost is low, and they have good economic benefits [16,17,18,19,20,21,22]. The open 3D frameworks containing (MO_4_)^n−^ polyanions instead of the smaller O^2−^ ions may allow for fast Li^+^ ion conduction.

In accordance with the aforementioned analysis, MnWO₄@C composites are successfully synthesized in this study via a solvothermal reaction. Surprisingly, the MnWO_4_@C electrode shows excellent electrochemical performance with a high capacity of 795 mAh g^−1^ at 200 mA g^−1^. And the MnWO_4_@C electrode also shows a superior capacity retention at various current densities from 100 to 5000 mA g^−1^. While the electrolyte additive can further improve the long-cycle stability of the MnWO_4_@C electrode, at 1000 mA g^−1^, the capacity of the MnWO_4_@C-FEC electrode is maintained at 718 mAh g^−1^ after 400 cycles.

## 2. Experimental Section

### 2.1. Materials Preparation

#### 2.1.1. Materials

Manganese chloride tetrahydrate (MnCl_2_·4H_2_O, AR, Macklin, Shanghai, China), Sodium tungstate dihydrate (Na_2_WO_4_·2H_2_O, Xi long hua gong, China), Urea (CH_4_N_2_O, Aladdin, China), Dopamine hydrochloride (PDA) (C_8_H_11_NO_2_·HCl, Aladdin, Shanghai, China), and Tris(hydroxymethyl)aminomethane (C_4_H_11_NO_3_, Aladdin, China) were used as the raw materials. All the chemicals and solvents were used as received without any further purification.

#### 2.1.2. Synthesis of MnWO_4_

As a typical preparing process, 0.3 mmol of MnCl_4_·4H_2_O and 0.3 mmol Na_2_WO_4_·2H_2_O were dissolved in 30 mL of deionized water (DIW) to form a solution A. After stirring for 30 min, 9 mmol of urea was added into solution A and stirred for another 30 min. Then, the mixture was transferred into a 50 mL polytetrafluoroethylene reactor, which was sealed and heated at 200 °C for 12 h. After cooling to the room temperature, the brown precipitate was collected and washed several times with DIW and then dried in a vacuum oven at 60 °C for 24 h. MnWO_4_ was obtained by annealing the intermediates under an air atmosphere at 450 °C for 12 h with a ramping rate of 2 °C min^−1^.

#### 2.1.3. Synthesis of MnWO_4_@C

We completely dispersed 100 mg of MnWO_4_ into 200 mL of Tris-buffer solution. Then, we added 60 mg of PDA to the above mixture and stirred for 12 h. The obtained product was cleaned and centrifuged multiple times using DIW, then dried at 60 °C, and finally baked in a vacuum furnace for 12 h (Figure 1).

#### 2.1.4. Material Characterization

A Bruker D8 Focus power type X-ray diffractometer, Cu Kα rays with a wavelength of 0.154056 nm, was used as the radiation source, and X-ray diffraction analysis was performed on the prepared samples at a scanning rate of 5°/min within the 2*θ* angle range of 10–80° to study their crystalline phases. Detailed observations and analyses of the appearance and microstructure of the synthesized product were conducted using a field emission scanning electron microscope (FE-SEM, model Hitachi S-4800, Hitachi, Tokyo, Japan) under a set acceleration voltage of 10 kV. A transmission electron microscope (TEM, model Tecnai G2, Hillsboro, OR, USA) was used to conduct an in-depth microstructure analysis of the sample under an electric field emission gun with an operating voltage of 200 kV. Then, we evaluated the surface chemical state and elemental composition of Al Kα source samples under X-ray photoelectron spectroscopy (XPS) analysis technology. In addition, the adsorption/desorption isotherm data of nitrogen were obtained using an ASAP 2010 trace adsorption analyzer under a temperature condition of −196 °C, combined with a porosity analysis function. The specific surface area of the sample was accurately calculated using the Brunauer–Emmett–Teller (BET) method. 

#### 2.1.5. Electrochemical Measurements

In the glove box, button type batteries were assembled, and the performance of the experimental batteries was tested. In order to prepare the working electrode, the prepared sample, acetylene black, and carboxymethyl cellulose (CMC) were mixed in a mass ratio of 80:10:10 and dissolved in an appropriate amount of deionized water. Sufficient stirring was used to ensure the formation of a uniform and consistent slurry. Then, we evenly coated the prepared slurry on the surface of the copper foil and then placed the copper foil coated with the slurry in a vacuum oven at 60 °C for 12 h of drying treatment. The average mass load of the active substance layer on the copper foil was controlled at 1.0–1.3 mg cm^−2^. The battery uses lithium foil as the counter electrode/reference electrode and a Celgard 2400 membrane as the separator, and the electrolyte was dissolved in a mixed solvent of ethylene carbonate (EC) and diethyl carbonate (DEC) (volume ratio 1:2) to form a complete half-cell system. For the anode testing for MnWO_4_@C, we added a mass percentage of 2% (wt) fluoroethylene carbonate (FEC) to the electrolyte to observe its impact on the testing process and results. We conducted a detailed evaluation and testing of the electrochemical performance of the battery samples using a programmable battery tester (LAND CT2001A) at room temperature. Cyclic voltammogram (CV) and electrochemical impedance spectroscopy (EIS) measurements were carried out on a Bio-Logic VMP3 electrochemical workstation.

## 3. Result and Discussion

### 3.1. The Characterization of MnWO_4_ and MnWO_4_@C

Figure 2a gives the morphology of MnWO_4_, which shows the shape of the nanorod with a diameter of about 20 nm, and the lengths of most nanorods are less than 200 nm. The TEM image in Figure 2b shows the hierarchically distributed nanorod-like structure more clearly. Figure 2c,d shows that there are plenty of micropores on the surface of nanorods, which can be used as effective channels for electrolyte transformation. The HRTEM image of Figure 2e clearly shows the lattice fringe (002) of the crystal plane of MnWO_4_, whose lattice spacing is 2.497 Å. The elements Mn, W, and O are evenly distributed, as can be seen from the EDX images in Figure 2f–j [23,24,25,26,27,28].

Figure 3 is TEM and EDX images of MnWO_4_@C. MnWO_4_@C also has the nanorod structure similar to MnWO_4_, as seen in Figure 3a. The nanorods are coated with a ~5.1 nm-thick carbon layer, which is derived from dopamine, as Figure 3b shows. The EDX analysis indicates that the elements Mn, W, O, and N are uniformly distributed in MnWO_4_@C nanorods. The element N comes entirely from the carbon layer-derived dopamine. Therefore, the range of the element N corresponds to the distribution of the carbon layer. The fuzzy distribution of the element C is caused by the carbon-supporting film in the testing process.

The composition of obtained materials was determined by Powder X-ray diffractometer (XRD), as shown in Figure 4a. From the XRD pattern, it can be seen that the patterns of MnWO_4_ and MnWO_4_@C are very similar. And both of them are corresponding to the Standards Card (PDF#13-0434, a = 4.824 Å, b = 5.75 Å, c = 4.99 Å) of MnWO_4_ very well. The absence of extraneous peaks confirms the high purity of the obtained sample [29,30,31]. Moreover, compared with the pure MnWO_4_, the inconspicuous protrusions around 21° proves a successful introduction of amorphous carbon into MnWO_4_@C. X-ray photoelectron spectroscopy (XPS) was used to analyze the chemical valence of elements in the sample. The peaks at 641.1 and 652.0 eV corresponded to the binding energy of Mn 2P_3/2_ and 2P_1/2_, respectively (Figure 4b). This shows that Mn is in the +2 oxidation state. The peaks at 33.6 and 36.2 eV are in accordance with the W 4f_7/2_ and 4f_5/2_, which indicates W is in the +6 oxidation state (Figure 4c). In the C 1s spectrum (Appendix A) [32], the signals at 284.5 eV, 285.4, and 288.5 eV correspond to the C=C, C-N (or C-C), and C=O configurations, respectively [25,33,34,35]. N_2_ adsorption/desorption measurements was used to determine the specific surface area and porous characteristics of MnWO_4_@C (Figure 4d,e: the latter is nested within Figure 4d). The specific surface area of MnWO_4_@C composite powder is 60.77 m^2^ g^−1^ from the type-III isotherm with a hysteresis loop. The average pore volume and average pore size are 0.18 cm^3^ g^−1^, respectively. The specific surface area of MnWO_4_ composite powder is 35.59 m^2^ g^−1^ from the type-III isotherm with a hysteresis loop (Appendix A) [36]. The increase in specific surface area is caused by the carbon shells derived from dopamine.

### 3.2. The Behavior of MnWO_4_ and MnWO_4_@C on Li^+^ Storage 

In order to evaluate the lithium storage capacity of MnWO_4_ and MnWO_4_@C, a lithium metal anode was employed as a symmetrical electrode to assemble a button battery. The MnWO_4_@C electrodes deliver an excellent capacity of 795 mAh g^−1^ at 200 mA g^−1^ after initial 100 cycles. This value is much higher than 340 mAh g^−1^ of bare MnWO_4_, which showed a spanning decay and 60% capacity retention after only 100 cycles, as shown in Figure 5a. This outstanding performance is also seen in rate performances (Figure 5b). MnWO_4_@C electrodes show superior capacity retention at various current densities from 100 to 5000 mA g^−1^. The average capacities of 875, 815, 687, 587, 447, and 287 mAh g^−1^ are obtained for MnWO_4_@C at 100, 200 500, 1000, 2000, and 5000 mA g^−1^, respectively; by contrast, 697, 408, 195, 102, 43, and 11 mAh g^−1^ were obtained for MnWO_4_. Even at high current densities of 2000 and 5000 mA g^−1^, its capacity retention can still be maintained at 51% and 32.8%. As the current densities return from 5000 to 100 mA g^−1^, the specific capacity comes back to 287, 500, 640, 735, 827, and 915 mAh g^−1^, indicating a good stability of MnWO_4_@C. Figure 5c shows the corresponding galvanostatic discharge–charge curve at different current densities of MnWO_4_@C. The cyclic voltammetry also certificates the stable cycling performance of MnWO_4_@C in a half-cell of LIBs at a scan rate of 0.1 mV s^−1^ in the voltage range from 0.01 to 3.0 V, as shown in Figure 5d. There are obviously differences between the first discharge curve and the subsequent cycle. In the first cycle, the peak at 1.2 V originates from the reduction of Mn^2+^ to Mn, and the reduced peak locked at 0.35 V corresponds to W^6+^ to W, as shown in Equation (1). In the subsequent anodic scan, there are two distinct peaks at 1.35 and 2.0 V, which corresponded to the transformation of Mn and W into MnO and WO_3_, respectively [37,38], as shown in Equation (2). The reaction process is as follows [39,40]:

Discharge: MnWO_4_ + 8Li^+^ +8e^−^ → Mn + W + 4Li_2_O (1)

Charge:Mn + W + 4Li_2_O → MnO + WO_3_ +8Li^+^ + 8e^−^
(2)

In the subsequent cycles, a pair of reduction peaks at 1.3 and 0.5 V in the cathodic scan and a pair of oxidation peaks at 1.3 and 2.0 V in the anodic scan were found. In subsequent loops, the oxidation peak at 2.0 V (peak I) gradually disappeared with the increase in the number of cycles, which proved the irreversibility of the W to WO_3_ response. In addition, the peak locked at 1.3 V does not change significantly in subsequent cycles, which proves that the electrochemical reaction is highly reversible. In the cathodic scan, the intensity of the peak II decreases gradually with the number of cycles increasing, which proved that the reversibility of the conversion reaction declined. This phenomenon is consistent with the change trend of the oxidation peak I. However, the intensity of the peak III becomes more and more obvious as the number of cycles increases, which proves that the reversibility of the reaction is getting much higher. This phenomenon corresponds to the change trend of the oxidation peak of 1.3 V in the anodic scan. When the current density is 1000 mA g^−1^, the capacity is maintained at 600 mAh g^−1^ after 200 cycles, which further proves the excellent electrochemical performance of MnWO_4_@C.

### 3.3. Kinetics Comparison of MnWO_4_ and MnWO_4_@C

The synthesis of N-doped carbon results in defects and pores being uniformly distributed throughout the material, creating a vast array of active sites and a homogeneous mesoporous structure, which enhances the electron conductivity and lithium-ion diffusion rate. Compared to MnWO_4_, MnWO_4_@C exhibits superior kinetic and exemplary electrochemical properties. Appendix A shows the Nyquist plots of MnWO_4_ and MnWO_4_@C [29]. The initial impedance semicircle observed in the high-frequency region predominantly represents the charge transfer resistance (*R_ct_*) at the electrode/electrolyte boundary. Subsequently, the second capacitive impedance arc emerges in the low-frequency region, attributable to the capacitive reactance (*R_c_*) of the adsorption layer generated on the electrode surface. The *R_ct_* value of MnWO_4_@C is 48.8 Ω, which is much lower than 64.7 Ω of MnWO_4_. In addition, the capacitive reactance of MnWO_4_@C (*R_c_* = 16.7 Ω) is also lower than that of bare MnWO_4_ (31.1 Ω), which implies MnWO_4_@C has a higher lithium-ion diffusion rate and better electrochemical performance. This proves that N-doped carbon not only relieves the stress of the lithium insertion process to a certain extent, maintaining the stability of the material structure but also forms a more stable SEI. The galvanostatic intermittent titration technique (GITT) test is also applied, which shows that MnWO_4_@C has better dynamic performance than MnWO_4_, as shown in Figure 6 and Appendix A [32] and Appendix A (tested at 0.1 A g^−1^, pulse time of 20 min, and relaxation time of 30 min). This is primarily attributable to the enhanced electrical conductivity of the active material, which is enabled by the N-doped carbon shells on the surface of MnWO_4_@C and effectively mitigates the volume effect during the cycling process, thereby facilitating the formation of a stable SEI film. Furthermore, surface imperfections (such as holes and cracks) can facilitate the formation of transport channels, which are conducive to the diffusion and transport of lithium ions. This, in turn, enhances the electrochemical cycling performance of the composites. The equation as follows [41]:$$D=\frac{4L2}{\pi \;\tau }(\frac{\Delta Es}{\Delta Et})2$$
in which L is the distance of Li^+^ diffusion, (equal to the electrode thickness, cm), τ stands for the relaxation time (s), ΔEs is the potential change via the current pulse, and ΔEt is the voltage change caused by constant current charging (discharging). The values of each parameter are shown in Figure 6b. According to the above equation, the lithium-ion diffusion rates under different delithiation and lithiation states are calculated, as shown in Appendix A. In addition, the cycle performance is comparable to previously reported results regarding Mn-based materials for LIBs (Appendix A).

As shown in Figure 7a, in order to delve deeper into the details of the charge storage process, we conducted cyclic voltammetry (CV) measurements and analyzed the correlation between the peak current (i) and the scanning rate (v) based on the power law i = av^b^. This analysis aims to reveal how the charge storage mechanism changes at different scanning rates, thereby providing a deeper understanding of the performance of a battery or electrode materials. Particularly, when the value of b is equal to 0.5, it indicates a semi-infinite linear diffusion control process, which means that the charge transfer is mainly limited by the diffusion rate of ions inside the electrode material; when the value of b reaches 1, it indicates that the process is surface controlled or has capacitive characteristics—that is, the storage and release of charges mainly occur on the surface or near the surface area of the electrode material, and the speed is very fast, almost not limited by diffusion. By plotting log i versus log v, the b values of the three redox peaks (peak 1, peak 2, and peak 3) are determined to be 0.74, 0.7, 1, and 0.79 (Figure 7b), indicating a dominant capacitive behavior of MnWO_4_@C during the electrochemical reaction. The equation i = k_1_v + k_2_v^1/2^ can be used to accurately quantify the proportion of capacitive contribution (k_1_v) and diffusive contribution (k_2_v^1/2^) in the total current. To determine the constant values of k1 and k2 at a specific potential, a simulation and fitting can be performed by plotting the relationship between iv^−1/2^ and v^1/2^. This method helps to separate and quantify these two different charge transfer mechanisms. The Figure 7c,d summarizes detailed contribution of capacitive behavior at various scan rates. The contributions are 62.98%, 74.67%, 79.12%, 81.82%, 83.62%, 84.33%, 86.83%, and 90.31% at the scan rates of 0.1, 0.3, 0.5, 0.7, 0.9, 1.0, 1.5, and 3.0 mV s^−1^, respectively. The experimental results show that, among the total capacity of MnWO_4_@C, capacitive charge storage accounts for a significant proportion. The enhanced charge transfer performance and fast capacitive charge storage capacity of the carbon shell in MnWO_4_@C are mainly attributed to its unique characteristics, including the rich amorphous structure, porous structure, and large specific surface area. In contrast, the contributions of MnWO_4_ are 29.6%, 42.1%, 48.4%, 52.6%, 55.8%, 57.1%, 62.0%, and 69.7% at the scan rates of 0.1, 0.3, 0.5, 0.7, 0.9, 1.0, 1.5, and 3.0 mV s^−1^, respectively, as Appendix A shows [42].

### 3.4. The Influence of an Electrolyte Additive

Fluoroethylene carbonate (FEC) is employed as a film-forming additive in electrolytes, whereby it forms stable complexes with lithium ions. This complexation constrains the migration of lithium ions, which in turn inhibits the occurrence of the reduction reaction of the boundary electrolyte, thus improving the cycle life of the battery. Concurrently, the reaction between FEC and lithium ions at the interface generates a dense film, which restricts the diffusion of solvent molecules and lithium ions in the electrolyte to the electrode surface, thereby reducing the self-discharge phenomenon of the battery [43,44,45,46]. Therefore, 2 wt% FEC was added in the electrolyte as an additive on the MnWO_4_@C electrode. The outstanding performance is represented in rate performances (Figure 8a). MnWO_4_@C-FEC electrodes shows high-capacity retention at various current densities from 100 to 5000 mA g^−1^. The average capacities of 878, 831, 758, 704, 611, and 458 mAh g^−1^ are obtained for MnWO_4_@-FEC at the current densities of 100, 200 500, 1000, 2000, and 5000 mA g^−1^, respectively, which are higher than MnWO_4_@C. As the current densities return from 5000 to 100 mA g^−1^, the specific capacity comes back to 678, 798, 824, 938, and 1017 mAh g^−1^, indicating more stability of MnWO_4_@C-FEC. Figure 8b shows the corresponding galvanostatic discharge–charge curve at different current densities of MnWO_4_@C-FEC, which is similar to the MnWO_4_@C anode. The capacity is maintained at 718 mAh g^−1^ at 1000 mA g^−1^ after 400 cycles, which further proves the superior electrochemical performance after adding FEC. However, the capacity of the MnWO_4_@C anode began to decrease after 200 cycles.

## 4. Conclusions

Through an efficient hydrothermal synthesis method, we have successfully prepared a compact nanocavity structure MnWO_4_@C nanorods. The formation mechanism of the nanocavity structure inside these nanorods is similar to the process of inside-out Ostwald ripening. The prepared MnWO_4_@C nanorods with dense nanocavities have been a substantial improvement in rate performance and specific capacity in LIB compared with MnWO_4_. On the one hand, the good electrochemical performance is mainly attributed to its unique nanocavity geometry and optimized conversion mechanism during the cycling process. These two factors jointly promote efficient electrode–electrolyte interface contact, establish a short-range electron transfer path, and provide sufficient lithium-ion storage sites during the insertion and extraction process of lithium ions. On the other hand, the standout electrochemical performance could be attributed to the carbon shell derived from dopamine. Moreover, both the transmission rate of lithium ions and the contribution of pseudo-capacitance increased because of the electroconductive and poriferous carbon shell. And the negative effects of volume expansion are also attenuated.

The carbon coating layer greatly improves the electrochemical properties of MnWO_4_ nanorods. However, with the cycle number increasing, the volume expansion of the electrode becomes more and more serious, and then the structure of the material and electrode are totally destroyed. Therefore, the electrolyte additive can form a stable SEI film and effectively restrain the capacity loss caused by the damage to the electrode structure in the long cycle process.

## Figures and Tables

**Figure 1 materials-17-04682-f001:**
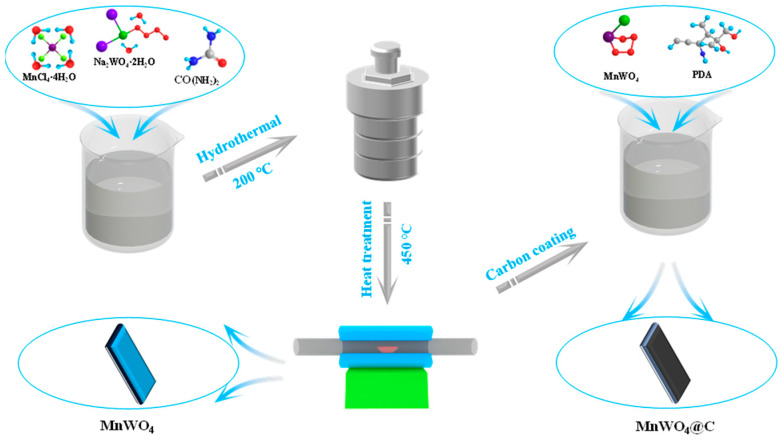
Scheme of the preparation process.

**Figure 2 materials-17-04682-f002:**
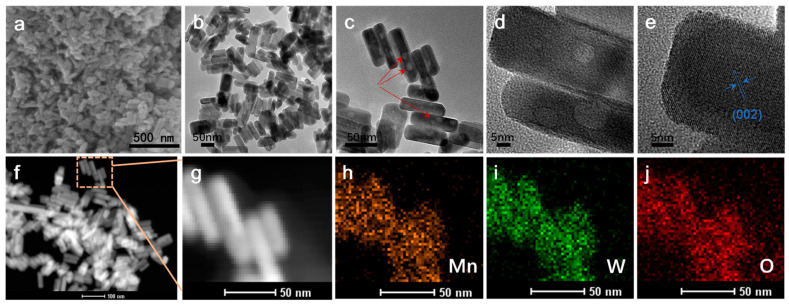
(**a**) SEM of MnWO_4_; (**b**–**e**) TEM and HRTEM of MnWO_4_; (**f**–**j**) EDX of MnWO_4_.

**Figure 3 materials-17-04682-f003:**
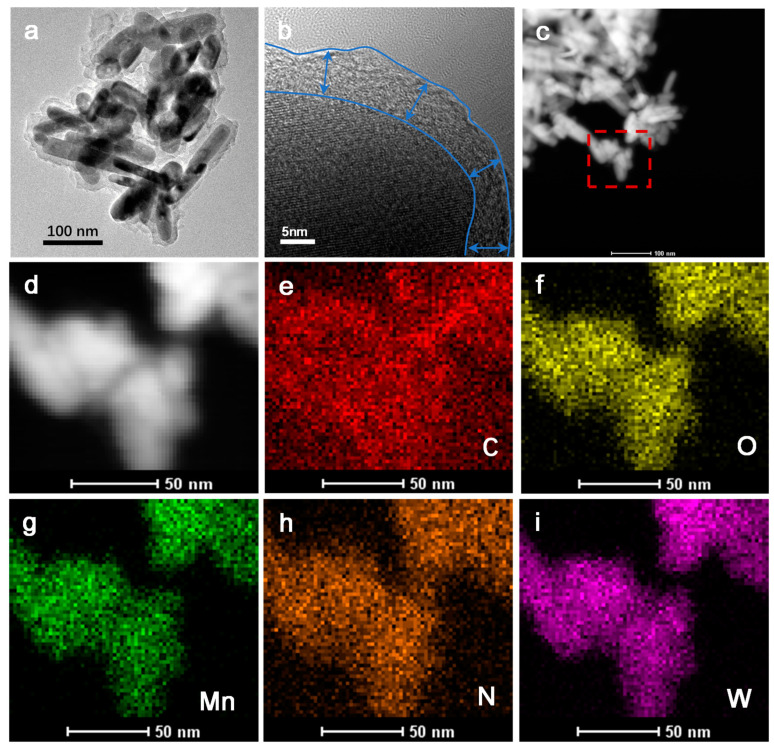
(**a**) TEM of MnWO_4_@C; (**b**) HRTEM of MnWO_4_@C; (**c**–**i**) EDX of MnWO_4_@C.

**Figure 4 materials-17-04682-f004:**
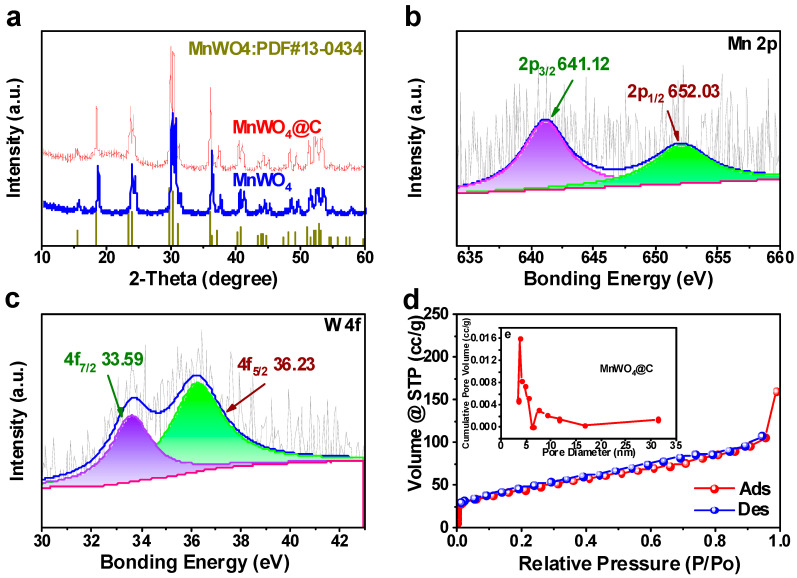
Physical characterization of MnWO_4_@C. (**a**) XRD patterns of MnWO_4_ and MnWO_4_@C. XPS spectra of the (**b**) Mn2p, (**c**) W4f, (**d**) N_2_ adsorption and desorption isotherms, and (**e**) pore size distribution of MnWO_4_@C.

**Figure 5 materials-17-04682-f005:**
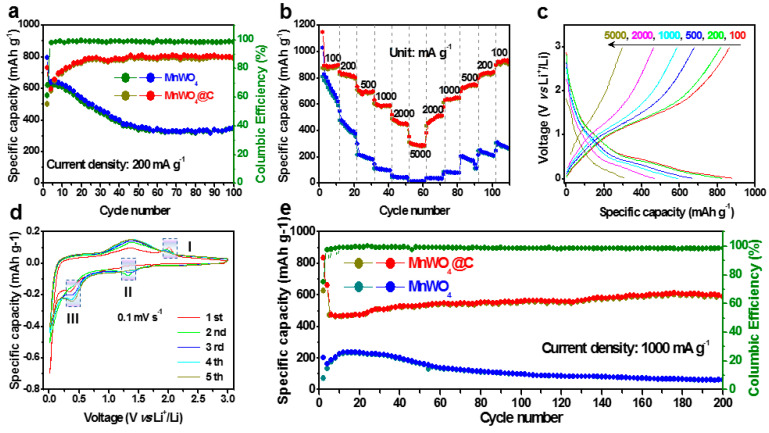
Electrochemical characteristics of MnWO_4_ and MnWO_4_@C. (**a**) Comparative cycle performances of MnWO_4_ and MnWO_4_@C at a current density of 200 mA g^−1^. (**b**) Rate capacities of MnWO_4_ and MnWO_4_@C at current densities of 100, 200, 500, 1000, 2000, and 5000 mA g^−1^. (**c**) galvanostatic discharge–charge curves of MnWO_4_@C electrodes at 0.1–5 A g^−1^. (**d**) CV curves of MnWO_4_@C electrodes at a scan rate of 0.1 mV s^−1^. (**e**) the galvanostatic discharge–charge cycling performances and the coulombic efficiency of MnWO_4_ and MnWO_4_@C electrodes at a current density of 1000 mA g^−1^.

**Figure 6 materials-17-04682-f006:**
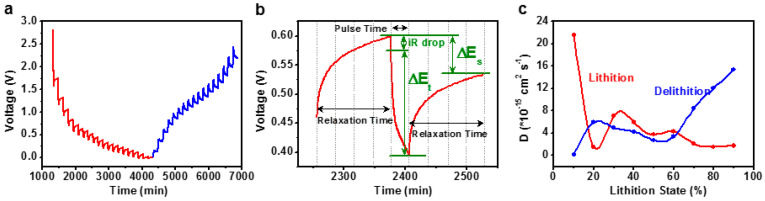
(**a**) GITT curves of the MnWO_4_@C electrode (discharge/charge state). (**b**) E vs. t profile for one GITT test. (**c**) D_Li_^+^ of MnWO_4_@C during the charge and discharge processes.

**Figure 7 materials-17-04682-f007:**
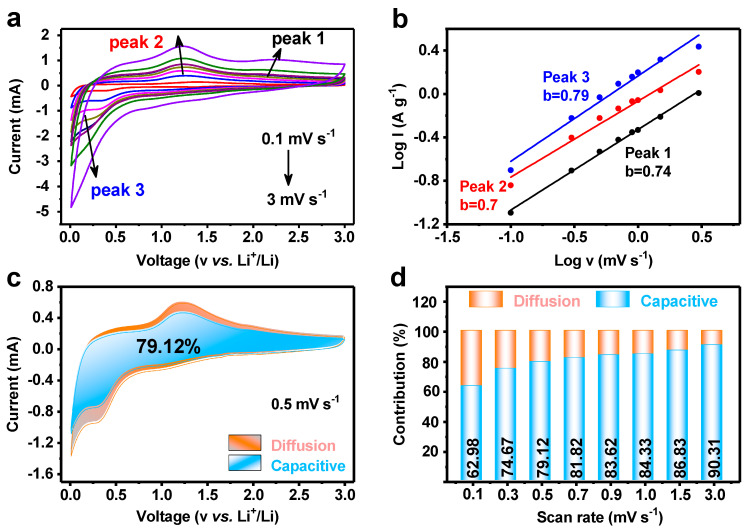
Kinetic analysis of MnWO_4_@C as LIB anodes. (**a**) CV curves of MnWO_4_@C at different scan rates, (**b**) log i versus log v plots at each redox peak, the contribution ratio of the capacitive and diffusion-controlled charge at (**c**) 0.5 mV s^−1^, and (**d**) different scan rates.

**Figure 8 materials-17-04682-f008:**
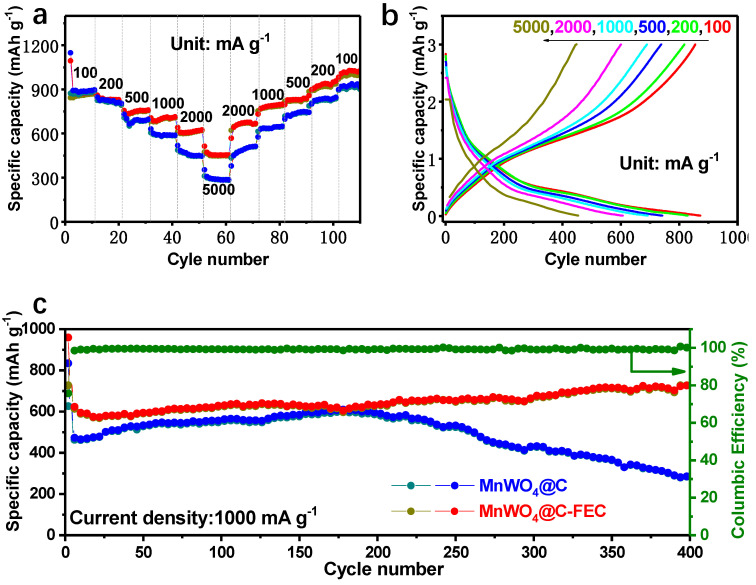
Electrochemical characteristics of MnWO_4_@C and MnWO_4_@C-FEC. (**a**) Comparative rate capacity of MnWO_4_@C and MnWO_4_@C-FEC at current densities of 100, 200, 500, 1000, 2000, and 5000 mA g^−1^. (**b**) galvanostatic discharge–charge curves of MnWO_4_@C-FEC electrodes at 0.1–5 A g^−1^. (**c**) the long cycle performance and coulombic efficiency of MnWO_4_@C and MnWO_4_@C-FEC electrodes at a current density of 1000 mA g^−1^.

## Data Availability

The authors confirm that the data supporting the findings of this study are available within the article [and its Appendix A].

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
