# Peer review of "Synergistically Boosting Li Storage Performance of MnWO4 Nanorods Anode via Carbon Coating and Additives"

_materials, 2024, doi:10.3390/ma17194682_

Round 1
Reviewer 1 Report
Comments and Suggestions for Authors
The work needs to be revised both in terms of content and language. The authors have presented a number of studies, but without a detailed analysis or indication of the added value of the proposed solution. For detailed comments, see below:
1. The text states that MnWO4@C has better kinetics than MnWO4, but does not explain in detail why this is the case. The role of N-doped carbon in enhancing kinetics should be elaborated upon, including its effect on electronic conductivity and lithium-ion diffusion.
2. The significance of having two semicircles in the high-frequency region should be discussed in more detail, specifically, how they relate to the solid electrolyte interphase (SEI) and charge-transfer resistance.
3. It is stated that MnWO4@C has a lower RSEI and RCT compared to MnWO4, but the impact of these lower resistances on overall electrochemical performance should be quantified or discussed in more detail.
4. The text mentions that N-doped carbon relieves the stress of the lithium insertion process, maintains material stability, and forms a more stable SEI. This is a critical point that needs further explanation. How does N-doping specifically contribute to these benefits? Are there structural or compositional analyses that support these claims?
5. The results of the GITT test indicate that MnWO4@C has better dynamic performance, but the explanation of why this is the case is lacking. The underlying mechanisms that lead to improved dynamic performance should be explained. How does the structure or composition of MnWO4@C contribute to its superior performance?
6. The text introduces fluoroethylene carbonate (FEC) as a film-forming additive, but does not provide a sufficient background on its significance and how it functions to improve battery performance. A brief explanation of the mechanism by which FEC enhances the SEI (solid electrolyte interphase) quality shoul be added.
7. Figure 8b shows the galvanostatic discharge–charge curves of MnWO4@C-FEC, which are similar to those of MnWO4@C. However, the significance of this similarity is not discussed. Explain how these curves support the claim of improved performance with the FEC additive.
8. The text briefly mentions that the capacity of MnWO4@C-FEC is maintained at 718 mAh g^-1 at 1000 mA g^-1 after 400 cycles, while MnWO4@C starts to decrease after 200 cycles. This is a crucial point that should be elaborated on. The authors should explain why the addition of FEC results in such a significant improvement in long-term cycling stability.
Comments on the Quality of English LanguageThe authors should correct for grammatical errors and improve sentence structure for better readability. For example “Figure 2a shows the morphology of MnWO4, revealing nanorods with a diameter of about 20 nm and lengths mostly less than 200 nm” instead “Figure 2a gives the morphology of MnWO4, it shows a shape of nanorod with a di- ameter of about 20 nm, and the length of most nanorods less than 200 nm”. Another example "In order to evaluate the lithium storage capacity of MnWO4 and MnWO4@C, lithium metal anode was employed as a symmetrical electrode to assemble a button battery." - This sentence is unclear. It should specify that a lithium metal anode was used in a half-cell configuration to test the MnWO4 and MnWO4@C electrodes in a button cell battery.
Author Response
Comments: The work needs to be revised both in terms of content and language. The authors have presented a number of studies, but without a detailed analysis or indication of the added value of the proposed solution. For detailed comments, see below:
Q1. The text states that MnWO4@C has better kinetics than MnWO4, but does not explain in detail why this is the case. The role of N-doped carbon in enhancing kinetics should be elaborated upon, including its effect on electronic conductivity and lithium-ion diffusion.
Reply: Thank you very much for your kind suggestion. The synthesis of N-doped carbon results in defects and pores being uniformly distributed throughout the material, creating a vast array of active sites and a homogeneous mesoporous structure, which enhances the electron conductivity and lithium-ion diffusion rate. Compared to MnWO4, MnWO4@C exhibits superior kinetic and exemplary electrochemical properties. More concrete details and analysis have been added in the revised manuscript. (Page 7, Lines 16-20)
Q2. The significance of having two semicircles in the high-frequency region should be discussed in more detail, specifically, how they relate to the solid electrolyte interphase (SEI) and charge-transfer resistance.
Reply: Thank you for your kind reminder. It is extremely essential to supplement a set of detail. The initial impedance semicircle observed in the high-frequency region predominantly represents the charge transfer resistance (Rct) at the electrode/electrolyte boundary. Subsequently, the second capacitive impedance arc emerges in the low-frequency region, attributable to the capacitive reactance (Rc) of the adsorption layer generated on the electrode surface. The added detailed analysis have been supplemented in the revised manuscript. (Page 7, Lines 20-28)
Q3. It is stated that MnWO4@C has a lower RSEI and RCT compared to MnWO4, but the impact of these lower resistances on overall electrochemical performance should be quantified or discussed in more detail.
Reply: Thank you for your good question. The Rct value of MnWO4@C is 48.8 Ω, which is much lower than the 64.7 Ω of MnWO4. In addition, the capacitive reactance of MnWO4@C (Rc = 16.7 Ω) is also lower than that of bare MnWO4 (31.1 Ω), which implies the MnWO4@C has a higher lithium-ion diffusion rate and better electrochemical performance. More concrete details and analysis have been added in the revised manuscript. (Page 7, Lines 25-28)
Q4. The text mentions that N-doped carbon relieves the stress of the lithium insertion process, maintains material stability, and forms a more stable SEI. This is a critical point that needs further explanation. How does N-doping specifically contribute to these benefits? Are there structural or compositional analyses that support these claims?
Reply: Thank you for your careful review. The incorporation of N-doped carbon shells has been demonstrated to enhance the electrical conductivity of the active material, thereby mitigating its volume effect during the cycling process. This, in turn, facilitates the formation of a stable SEI film, which, in turn, improves the cycling stability of the battery. Furthermore, the surface imperfections of N-doped carbon shells (such as holes and cracks) can serve as the initial point for SEI film formation. These imperfections provide active sites that facilitate the adsorption and aggregation of electrolyte decomposition products, ultimately leading to the formation of the SEI film. Furthermore, the surface defects can provide additional reaction sites, thereby promoting the growth and stability of SEI film. Consequently, N-doped carbon shells with excellent chemical stability and corrosion resistance can safeguard the integrity of the SEI film and extend the cycle life of the battery.
Q5. The results of the GITT test indicate that MnWO4@C has better dynamic performance, but the explanation of why this is the case is lacking. The underlying mechanisms that lead to improved dynamic performance should be explained. How does the structure or composition of MnWO4@C contribute to its superior performance?
Reply: Thank you for your kind suggestion. This is primarily attributable to the enhanced electrical conductivity of the active material, which is enabled by the N-doped carbon shells on the surface of MnWO4@C and effectively mitigates the volume effect during the cycling process, thereby facilitating the formation of a stable SEI film. Furthermore, surface imperfections (such as holes and cracks) can facilitate the formation of transport channels, which are conducive to the diffusion and transport of lithium-ion. This, in turn, enhances the electrochemical cycling performance of the composites. More concrete details and analysis have been added in the revised manuscript. (Page 7, Lines 33-40)
Q6. The text introduces fluoroethylene carbonate (FEC) as a film-forming additive, but does not provide a sufficient background on its significance and how it functions to improve battery performance. A brief explanation of the mechanism by which FEC enhances the SEI (solid electrolyte interphase) quality should be added.
Reply: Thank you for your good question and kind suggestion. Fluoroethylene carbonate (FEC) is employed as a film-forming additive in electrolytes, whereby it forms stable complexes with lithium ions. This complexation constrains the migration of lithium ions, which in turn inhibits the occurrence of the reduction reaction of the boundary electrolyte, thus improving the cycle life of the battery. Concurrently, the reaction between FEC and lithium ions at the interface generates a dense film, which restricts the diffusion of solvent molecules and lithium ions in the electrolyte to the electrode surface, thereby reducing the self-discharge phenomenon of the battery. The specific discussion has been supplemented in the revised manuscript. (Page 9, Lines 13-19)
Q7. Figure 8b shows the galvanostatic discharge-charge curves of MnWO4@C-FEC, which are similar to those of MnWO4@C. However, the significance of this similarity is not discussed. Explain how these curves support the claim of improved performance with the FEC additive.
Reply: Thank you for your good question and kind suggestion. The incorporation of fluorine ethylene carbonate (FEC) as a film-forming additive in the electrolyte does not result in any alteration to the charging and discharging mechanisms of such batteries, it merely forms a stable complex compound with lithium ions. This complexation constrains the migration of lithium ions, which in turn inhibits the occurrence of the reduction reaction of the boundary electrolyte, thus improving the cycle life of the battery. Concurrently, the reaction between FEC and lithium ions at the interface generates a dense film, which restricts the diffusion of solvent molecules and lithium ions in the electrolyte to the electrode surface, thereby reducing the self-discharge phenomenon of the battery.
Q8. The text briefly mentions that the capacity of MnWO4@C-FEC is maintained at 718 mAh g-1 at 1000 mA g-1 after 400 cycles, while MnWO4@C starts to decrease after 200 cycles. This is a crucial point that should be elaborated on. The authors should explain why the addition of FEC results in such a significant improvement in long-term cycling stability.
Reply: Thank you for your good question and kind suggestion. Fluoroethylene carbonate (FEC) is employed as a film-forming additive in electrolytes, whereby it forms stable complexes with lithium ions. This complexation constrains the migration of lithium ions, which in turn inhibits the occurrence of the reduction reaction of the boundary electrolyte, thus improving the cycle life of the battery. Concurrently, the reaction between FEC and lithium ions at the interface generates a dense film, which restricts the diffusion of solvent molecules and lithium ions in the electrolyte to the electrode surface, thereby reducing the self-discharge phenomenon of the battery. The specific discussion has been supplemented in the revised manuscript. (Page 9, Lines 13-19)

Reviewer 2 Report
Comments and Suggestions for Authors
This manuscript presents an alternative anode material for Li-ion batteries. It is complete, the results are extensive and well-discussed, and the conclusions are sound. Please revise the introduction. The last paragraph of section 1 is incomplete, and understanding it is rather difficult. Also, in the experimental section, revise the writing of section 2.1.1 (Materials).
Comments on the Quality of English LanguageA revision by a native speaker would improve the quality and readability of the manuscript. This reviewer strongly suggests doing it.
Author Response
Comments: This manuscript presents an alternative anode material for Li-ion batteries. It is complete, the results are extensive and well-discussed, and the conclusions are sound. Please revise the introduction. The last paragraph of section 1 is incomplete, and understanding it is rather difficult. Also, in the experimental section, revise the writing of section 2.1.1 (Materials).
Reply: Thank you for your kind suggestion. The specific modifications have been supplemented in the revised manuscript. (Page 1, Lines 39-40; Page 2, Lines 9-10)

Reviewer 3 Report
Comments and Suggestions for Authors
Feng et.al reported the “Synergistically Boosting Li Storage Performance of MnWO4 nanorods anode via Carbon Coating and Additives” for energy storage purposes. The authors prepared MnWO4-based anodes and modified them with carbon/additives to enhance the performance of lithium-ion batteries. The manuscripts lack in technical aspects and need significant revision.
1. How authors concluded 5.1 nm carbon coating layer formed on nano rods. Also, dopamine-assisted carbon coating methods were used. What are the critical parameters and experimental conditions utilized to achieve the controlled thickness of the carbon layer?
2. Why surface are increased after being treated with dopamine? Understandably, dopamine converts into carbon by heating. However, as mentioned previously it has only a 5.1 nm coating of carbon layer. Will such a fine thin coating change the surface area? The authors need to provide more discussion.
3. What is the coluombic efficiency of with and without carbon-coated MnWO4 during charge/discharge? It is found that carbon coating enhances the surface and can enhance electrolyte decomposition which leads to less coulombic efficiency. What is the author’s opinion on this?
4. The authors used GITT measurement to estimate the diffusion coefficient values. From the figure, it is found that diffusion coefficient values not much changed even after carbon coating. Also, the values are in the order 10-15 cm2/s and look very low compared to other materials in the literature. It is better to provide a comparison table of other anodes with details such as capacity, cycle life, C-rate, and diffusion coefficient values.
5. It is better to provide detailed mechanisms undergoing during the lithium charge/discharge process. Are it proposed materials following intercalation or surface absorption or conversion type mechanism during the lithiation/delithiation process?

English revision is required in many places for better understanding.
Author Response
Q1. How authors concluded 5.1 nm carbon coating layer formed on nanorods. Also, dopamine-assisted carbon coating methods were used. What are the critical parameters and experimental conditions utilized to achieve the controlled thickness of the carbon layer?
Reply: Thank you very much for your kind issues. (1) As illustrated in Fig. 1, the encapsulated carbon layer is observed to have a thickness of approximately 5 nm. (2) Given the sensitivity of the self-polymerization reaction of dopamine to changes in pH, it is essential that the coating is carried out in a buffer solution with a pH of 8.5. Furthermore, the thickness of the carbon layer can be regulated by modifying the concentration of the dopamine solution and the duration of the capping process.
Fig 1. TEM of MnWO4@C.
Q2. Why surface are increased after being treated with dopamine? Understandably, dopamine converts into carbon by heating. However, as mentioned previously it has only a 5.1 nm coating of carbon layer. Will such a fine thin coating change the surface area? The authors need to provide more discussion.
Reply: We greatly thank the reviewer’s kind reminder. After heat treatment, dopamine is converted into a carbon material which has a distinct porous structure and a significant increase in specific surface area compared to MnWO4. It can also be seen from Fig. 2 and Fig. 3 that the specific surface area of MnWO4@C is significantly higher than that of MnWO4.
Fig 2. N2 adsorption and desorption isotherms and pore size distribution of MnWO4@C.
Fig 3. N2 adsorption and desorption isotherms of MnWO4.
Q3. What is the coluombic efficiency of with and without carbon-coated MnWO4 during charge/discharge? It is found that carbon coating enhances the surface and can enhance electrolyte decomposition which leads to less coulombic efficiency. What is the author’s opinion on this?
Reply: The initial coulombic efficiency of the MnWO4@C composite is modestly superior to that of MnWO4. The presence of excessive carbon material has been observed to have a detrimental impact on the first coulombic efficiency of the electrode material. However, in this work, a small portion of the carbon material has been shown to enhance both the electrical conductivity and the cycling reversibility of MnWO4. Although interfacial side reactions occur between the coated carbon material and the electrolyte, the low carbon content has been demonstrated to contribute to enhancing the initial coulombic efficiency to a certain extent.
Q4. The authors used GITT measurement to estimate the diffusion coefficient values. From the figure, it is found that diffusion coefficient values not much changed even after carbon coating. Also, the values are in the order 10-15 cm2/s and look very low compared to other materials in the literature. It is better to provide a comparison table of other anodes with details such as capacity, cycle life, C-rate, and diffusion coefficient values.
Reply: Thank you for your careful review. The details of the comparison table for other anodes are shown below:
Table S3. Comparison of electrochemical performance in LIBs with previous works.
|
Samples |
Rate (C) |
Cycle number |
Capacity (mAh g-1) |
DLi+(cm2 s-1) |
Refs. |
|
MnWO4 nanobars (LIBs) |
0.1 |
160 |
600 |
|
S1 |
|
MnWO4 nanoparticles (LIBs) |
0.2 |
150 |
340 |
10-19-10-20 |
S2 |
|
MnWO4@MWCNTs (LIBs) |
0.2 |
30 |
425 |
|
S3 |
|
MnWO4@C (LIBs) |
0.1 |
100 |
1063 |
|
S4 |
|
F-doped nano-MnWO4 (LIBs) |
0.2 |
150 |
200 |
10-19-10-20 |
S5 |
|
MnWO4@C (LIBs) |
1 |
200 |
600 |
10-15 |
This work |
References
[S1] En Zhang, Zheng Xing, Ji Wang, Zhicheng Ju, Yitai Qian, Enhanced energy storage and rate performance induced by dense nanocavities inside MnWO4 nanobars, RSC Advances, 2 (2012) 6748-6751.
[S2] Wei Wang, Na Wu, Jin-Ming Zhou, Feng Li, Yu Wei, Tao-Hai Li, Xing-Long Wu, MnWO4 nanoparticles as advanced anodes for lithium-ion batteries: F-doped enhanced lithiation/delithiation reversibility and Li-storage properties, Nanoscale, 10 (2018) 6832-6836.
[S3] Hyun-Woo Shim, Ah-Hyeon Lim, Jae-Chan Kim, Gwang-Hee Lee, Dong-Wan Kim, Hydrothermal realization of a hierarchical, flowerlike MnWO4@MWCNTs nanocomposite with enhanced reversible Li storage as a new anode material, Chem. Asian J, 8 (2013) 2851-2858.
[S4] Ge Gao, Wei Dang, Huimin Wu, Guangxue Zhang, Chuanqi Feng, Synthesis of MnWO4@C as novel anode material for lithium ion battery, Journal of Materials Science: Materials in Electronics, 29 (2018) 12804-12812.
[S5] Jianyu Wei, Jinxiu Ma, Wei Wang, Taohai Li, Na Wu, Dabin Zhang, Study of the effect of F‑doping on lithium electrochemical behavior in MnWO4 anode nanomaterials, Journal of Inorganic and Organometallic Polymers and Materials, 31 (2021) 3175-3182.
The specific table has been supplemented in the revised manuscript. (Table S3, Supporting information)
Q5. It is better to provide detailed mechanisms undergoing during the lithium charge/discharge process. Are it proposed materials following intercalation or surface absorption or conversion type mechanism during the lithiation/delithiation process?
Reply: Thank you for your kind suggestion. The detailed mechanism of the lithium charging and discharging process is as follows:
Discharge:
MnWO4 + 8Li+ +8e- → Mn + W + 4Li2O (1)
Charge:
Mn + W + 4Li2O → MnO + WO3 + 8Li+ + 8e- (2)
It is evident that the MnWOâ‚„ material adheres to a conversion mechanism throughout the lithiation/delithiation process.
We sincerely thank you and all the reviewers for their valuable comments. We tried our best to improve the manuscript. These changes will not influence the content of the paper. We appreciate for your warm work earnestly, and hope that the correction will meet with approval. Once again, thank you very much for your comments.

Round 2
Reviewer 1 Report
Comments and Suggestions for Authors
I have no futher comments.
Reviewer 3 Report
Comments and Suggestions for Authors
The authors sufficiently addressed the reviewer's comments and included them in the revised manuscript. The manuscript can be considered for further process.